# "Each moon we come to weigh the pregnancy:" Exploring the experience of group antenatal care processes in Benin and their contributions to self-efficacy

Julie Niemczura Sutton[1], Fifamè Aubierge Eudoxie Kpatinvoh[2], Esther Firmine Cadja Dodo[2], Erin Go[3], Courtney Emerson[4], Catherine Dentinger[4], Stephanie Suhowatsky[5], Julie Buekens[1], Katherine Wolf[5], Manzidatou Alao[2], Marie Adeyemi Idohou[2], Faustin Onikpo[2], Cyriaque D. Affoukou[6], Aurore Ogouyèmi-Hounto[7], Julie R. Gutman[8], Peter J. Winch[9]*

1 U.S. President's Malaria Initiative Impact Malaria Project/MCD Global Health, Silver Spring, Maryland, United States of America, 2 U.S. President's Malaria Initiative Impact Malaria Project/MCD Global Health, Cotonou, Benin, 3 Rutgers University, New Brunswick, New Jersey, USA/Johns Hopkins Bloomberg School of Public Health, Baltimore, Maryland, United States of America, 4 U.S. President's Malaria Initiative, Malaria Branch, Division of Parasitic Diseases and Malaria, Centers for Disease Control and Prevention, Atlanta, Georgia, United States of America, 5 U.S. President's Malaria Initiative Impact Malaria Project/Jhpiego, Baltimore, Maryland, United States of America, 6 National Malaria Control Program, Ministry of Health, Cotonou, Benin, 7 Unité de Parasitologie, Faculté des Sciences de la Santé, Université d'Abomey Calavi, Cotonou, Benin, 8 Malaria Branch, Division of Parasitic Diseases and Malaria, Centers for Disease Control and Prevention, Atlanta, Georgia, United States of America, 9 U.S. President's Malaria Initiative Impact Malaria project/Department of International Health, Johns Hopkins Bloomberg School of Public Health, Baltimore, Maryland, United States of America

* pwinch@jhu.edu

## Abstract

Group antenatal care (G-ANC) is a model that brings pregnant women with similar estimated delivery dates together for clinical assessment and participatory learning in a supportive social context. A qualitative study was nested in a trial assessing the community impact of G-ANC on ANC retention and uptake of intermittent preventive treatment of malaria in pregnancy (IPTp) in Atlantique Department, Benin. This nested qualitative study assessed women's experience of G-ANC, and ways participation could foster self-efficacy to perform a variety of prevention and care-seeking behaviors. Ten semi-structured focus group discussions were conducted with 129 women who attended G-ANC; deductive thematic codes were informed by Bandura's four sources of efficacy expectations. Recently pregnant women's experiences with individual ANC versus G-ANC were assessed via household surveys. G-ANC participation proffered three sources of self-efficacy expectations: performance accomplishments, verbal persuasion, and vicarious experience. Among household survey respondents, 96% (134/140) of women who participated in G-ANC would prefer it over individual ANC for future pregnancies. While a higher proportion of G-ANC participants felt that the provider answered all their questions in a way they could understand, most women reported that not all their questions were answered, even

**Data availability statement:** Data have been uploaded to the CDC data repository at data. cdc.gov. The data analysis results from the relevant section of the endline household survey underlying the data tables in this article can be found at https://data.cdc.gov/Global-Health/Benin_GANC_BehavioralData/8592-cnvk/about_data in PDF. Also included is an English-language version of all behavioral questions used in the household survey, which was adapted from the Malaria Behavior Survey (MBS) and subsequently translated into French and Fongbe. For the study protocol, qualitative interview guides and data requests in other formats, you may contact Dr. Julie R. Gutman, Principal Investigator, United States Centers for Disease Control and Prevention (CDC). Email: fff2@cdc.gov.

**Funding:** This study was made possible by the generous support of the American people through the President's Malaria Initiative under the terms of United States Agency for International Development (USAID) contract 7200AA18C00014 (JNS, FAEK, EFCD, SS, JB, KW, MA, MAI, FO, CDA, AOH, PJW). The IDARE office of the Johns Hopkins University Bloomberg School of Public Health provided funding support to EG. USAID had a role in the decision to publish the manuscript. The findings and conclusions presented in this paper are those of the authors and do not necessarily reflect the views of USAID, CDC, or the United States Government.

**Competing interests:** The authors have declared that no competing interests exist.

in G-ANC. G-ANC processes fostering self-efficacy to overcome barriers to ANC attendance may have facilitated women's participation in G-ANC meetings as well as taking more doses of IPTp. Self-efficacy of pregnant women participating in G-ANC could be strengthened by providers addressing all participants' questions in a more complete and understandable way, contributing to more effective verbal persuasion. Other parallel processes during G-ANC need to be maintained to provide multiple sources of self-efficacy for health behaviors like timely care-seeking, pregnancy management, pregnancy self-care, and facility birth.

## Introduction

Group antenatal care (G-ANC) is a service delivery model in which pregnant women with similar due dates are seen together in a group. This model was designed to promote adoption of key behaviors to increase the probability of healthy pregnancy outcomes [1,2]. A 2022 G-ANC theory of change published by Grenier et al. places self-efficacy on the causal pathway to key behavior adoption [3]. Participatory education that providers deliver during G-ANC covers myriad health topics, helping women address barriers to successfully seek family planning, prevent or obtain treatment for malaria, take early action in response to danger signs during pregnancy, and deliver at a health facility. Recent studies indicate that G-ANC can increase knowledge, confidence, and empowerment [4–7], the likelihood of women having four or more antenatal care contacts (ANC4) [5,8,9], uptake of interventions for malaria in pregnancy [10], and postpartum family planning use [11,12]. Self-efficacy figures prominently among several factors affecting behavior adoption [13], yet the G-ANC processes that have the potential to increase self-efficacy have not been closely studied in low- and middle-income countries (see S1 Appendix for details of G-ANC processes and theorized contributions to self-efficacy).[3]

Albert Bandura defines self-efficacy as perceived capability or belief that one can successfully complete a specific task or behavior [13]. Theoretically, G-ANC processes (e.g., taking and recording vital signs of other participants, learning from modeling experiences of others similar to oneself [Fig 1 and Table 1]) should strengthen participants' self-efficacy by influencing Bandura's four sources of self-efficacy expectations: 1) performance accomplishments, 2) vicarious experience, 3) verbal persuasion, and 4) emotional/physiological arousal [13].

A nested qualitative study was conducted as part of a cluster randomized controlled trial (cRCT) assessing the impact of G-ANC on IPTp and ANC attendance in Atlantique Department, Benin [15]. The cRCT sought to ascertain whether G-ANC could be implemented at sufficient scale to measurably improve the quality and quantity of pregnant women's ANC contacts, and prevent malaria in pregnancy at the community level. The nested qualitative component assessing the feasibility and acceptability of G-ANC as a novel service delivery model was essential to achieve the parent study's key aims, insomuch as scaling the intervention depended on various factors that were best understood and evaluated

| Pregnant women recruited into G-ANC groups | | |
|---|---|---|

↓

| Participation in G-ANC meetings | | |
|---|---|---|
| **Sources of efficacy expectations**<br>• Performance Accomplishments<br>• Vicarious experiences<br>• Verbal persuasion | **Processes in G-ANC contributing to efficacy expectations**<br>• Taking vital signs and weight<br>• Taking antimalarial drugs<br>• Midwives sharing experiences of other women<br>• Women sharing own experiences<br>• Encouragement and feedback during G-ANC meetings, and later by phone<br>• Detailed instruction on behaviors | |

↓

| **Behaviors targeted in four types of self-efficacy** | • Emergency care-seeking<br>• Pregnancy management<br>• Pregnancy self-care<br>• Facility delivery |
|---|---|

↓

| **Behavioral Outcomes** | • Increased uptake of three or more doses of IPTp<br>• Increased ANC attendance: four or more times during pregnancy |
|---|---|

↓

| **Health Outcomes (theorized)** | • Reduced malaria in pregnancy and its consequences – low birthweight, pre-term birth<br>• Reduced maternal and perinatal mortality and morbidity |
|---|---|

**Fig 1. Processes, strategies and behavioral outcomes related to self-efficacy among pregnant women in group antenatal care.**

qualitatively. This secondary qualitative analysis has two aims: 1) To describe pregnant women's experience of participation in G-ANC to understand its acceptability in the Beninese context, in direct support of the parent study's key outcome of interest; 2) To go beyond the scope of the parent study to examine ways G-ANC might foster pregnant women's self-efficacy to carry out any number of desirable health behaviors, and relate them to Bandura's four sources of efficacy expectations [13].

## Methods

Quantitative and qualitative data collection took place from November 2, 2020 to December 15, 2022.

### Group antenatal care

From March 2021-September 2022, pregnant women attending individual ANC before 25 weeks' gestation were invited to join a group of 8–15 women with similar due dates for five monthly G-ANC meetings at their health facility instead of individual ANC. Before the start of each meeting, women took each other's blood pressure with a digital cuff and recorded

**Table 1. Processes in group antenatal care sessions influencing Bandura's four sources of efficacy expectations.**

| Source of efficacy expectations from Bandura 1977 [13] | Mode of induction for each source according to Bandura 1977 [13], 1982 [14] | Mode of induction for each source within G-ANC sessions |
|---|---|---|
| **Performance accomplishments** | Participant modeling<br>Performance desensitization<br>Performance exposure<br>Self-instructed performance | Successfully taking and recording vital signs<br>Successfully taking and recording weight<br>Successfully taking IPTp<br>Successfully trying different behaviors at home, and reporting back to others during G-ANC |
| **Vicarious experience** | Live modeling<br>Symbolic modeling | Observation of peers successfully taking vital signs<br>Observation of peers successfully taking IPTp<br>Sharing positive and negative experiences from previous pregnancies<br>Role-playing |
| **Verbal persuasion** | Suggestion<br>Exhortation<br>Self-instruction<br>Interpretive treatments | Encouragement and feedback for behaviors both during G-ANC sessions, and by phone afterward<br>Encouragement from providers and peers to return to G-ANC<br>Step-by-step instruction on how to carry out behaviors such as iron supplementation |
| **Emotional arousal and physiological states** | Relaxation, elimination of emotional reactions or anticipatory stress<br>Exposure to behaviors and desensitization to somatic responses derived from mastery experiences | Emotional arousal was not described in focus groups as a significant contributor to self-efficacy, likely because self-efficacy around the main behaviors was high at baseline and limited any physiological distress experienced |

this along with weight and temperature, sometimes assisted by a health aide. Every woman was individually examined by a provider in private, before or during the meeting. (See S2 Appendix).

After an opening ritual, a nurse or midwife trained in G-ANC facilitation introduced the meeting topic from the G-ANC manual and led a participatory session where women interacted as a group and supported each other. The five-meeting series covered nutrition, malaria prevention, identification of danger signs in pregnancy, family planning, and preparation for birth. Topics were harmonized with Benin's Ministry of Health recommendations; approximately one hour was allocated to each topic, compared to five minutes or less to cover all topics in an individual ANC visit. After covering the topic of the month, facilitators distributed sulfadoxine-pyrimethamine (SP) for IPTp under directly observed therapy. Each meeting ended with a closing song and dance.

## Qualitative methods - focus groups

From May-July 2022, two health facilities in each of Atlantique's three health zones were purposively selected, for a total of six out of 20 intervention sites, to include one high (>100 clients/month) and one low (<100 clients/month) ANC client volume site in each zone (see S3 Appendix). Site selection also considered differing G-ANC enrollment and retention rates (see S3 Appendix for characteristics of all intervention health facilities). A convenience sample of women who attended G-ANC more than twice during their most recent pregnancy were invited to join focus groups; separately, others who attended G-ANC only one or two times were invited. Ten focus groups were conducted by two trained interviewers in outdoor pavilions, open spaces, or health facility meeting rooms to understand women's experience in their most recent pregnancy and with G-ANC, prevention of malaria in pregnancy, and family planning (see S4 Appendix for the focus group discussion guide). In addition to the focus groups anticipated in the study protocol, three intervention sites had multiple focus groups because of a higher-than-expected participation rate among invited G-ANC participants. Two focus groups were added with male partners to explore themes that emerged during the pre-testing of qualitative research tools.

## Quantitative methods – household surveys

The parent study included baseline and endline household surveys in November-December 2020 and October-December 2022. Survey respondents were selected at random from a list of all women who had given birth in the preceding 12 months in one enumeration area per health facility catchment; thus women in the intervention and control arms were surveyed at endline regardless of whether they participated in G-ANC. The household survey asked questions about the parent study's main outcomes of interest, i.e., IPTp uptake and ANC attendance, as well as questions adapted from the Malaria Behavior Survey Women's Module [16] about ANC and perceived self-efficacy to attend ANC and take IPTp [17].

## Data management and analysis

Focus groups were conducted in Fongbe, recorded, and transcribed into French in a template based on the focus group discussion guide. Results were coded and summarized in a Google Form by EG and PJW in consultation with FAEK and EFCD, following an adapted framework analysis process [18,19]. An initial round of coding and data summarization was conducted in French by the qualitative researchers based in Benin. A second round of coding with specific focus on self-efficacy was done in English on translated transcripts. There was frequent return to the French transcripts to confirm the translation. Deductive thematic analysis assessed women's experiences and examined mechanisms through which participation in G-ANC promoted self-efficacy to adopt preventive and care-seeking behaviors based on Albert Bandura's self-efficacy framework [13]. While pre-determined codes for role-playing, feedback, hands-on experience, and encouragement were applied in relation to Bandura's four sources of self-efficacy (see Results), other aspects of G-ANC were identified as recurring themes. Searches for self-efficacy statements included: 1) perceived ability to carry out behaviors, specifically looking for "can" and not "will" statements because confidence and capability are predictors of intention, not equivalent to it [13]; 2) signs of confidence and perceived capability; and 3) signs of confidence and perceived capability."

Household survey data for the quantitative portion of the study were entered into the CommCare platform, exported, and analyzed with SAS version 9.4 (SAS Institute Inc., Cary, NC, USA) using a difference in differences approach, accounting for clustering and survey weighting, when comparing data collected at both baseline and endline. For analyses restricted to endline data only, we used proc surveyfreq in SAS V9.4 to account for clustering and weighting. Chi squared tests were used to assess statistical significance. Where any of the cells of a 2x2 table had less than 5 individuals, we used Fisher's exact test. Data analysis is described more fully in the main outcome paper [15].

## Ethical considerations

The Institutional Review Board of the Centers for Disease Control and Prevention (ID 7254) and the Ethics Committee of the Benin Ministry of Health (ID 3/MS/DRFMT/CNERS/SA) approved the protocol prior to implementation of study activities. All data collectors completed the CITI online course on human subjects research. Community sensitization ensured that local leaders and community members were informed about the study. Health zone officials and facility in-charges provided permission for study activities in each health center and village chiefs provided permission for household survey activities. Written informed consent was obtained from each person prior to their participation in G-ANC, surveys, or semi-structured interviews. Patients or the public were not involved in the design, conduct, reporting, or dissemination plans of this research. Additional information regarding the ethical, cultural, and scientific considerations specific to inclusivity in global research is included in the Supporting Information (S1 File).

## Results

Across 20 health facilities designated as intervention sites in the parent study, 2,516 women attended at least one G-ANC meeting. This number corresponded to 14% of all pregnant women at the intervention sites receiving G-ANC. The endline survey identified 140 of these participants out of a random sample of 1,280 households and found G-ANC

participants were 1.9 times more likely than those who received only individual ANC to have attended four or more ANC visits (p = 0.002) and 1.8 times more likely to have taken three or more IPTp doses (p = 0.004). Of 140 G-ANC participants invited to participate in focus group discussions — recruited separately from the 140 G-ANC participants identified during the endline survey — 129 participated. Nearly all focus group participants were married, averaged 26.7 years old, and had a mean of 2.8 children. The results of the secondary analysis that follow draw from both the endline household survey and ten qualitative focus group discussions.

**Acceptability of group care**

Women who attended G-ANC reported a positive experience with it, and participants in most focus groups said they preferred G-ANC over individual ANC based on their experience:

"Before, when I came [to the health center], I felt alone, I sat alone and when my turn came, I would go to the midwife, and she would examine me. But now, the atmosphere is different, you feel like family, you have sisters."

Women described receiving social support for their continued participation in G-ANC that helped them find solutions to such barriers to care as transportation costs, consultation fees, and hesitant spouses. They felt more confident advocating for themselves when faced with stressful situations.

"If you have a problem at home, you can tell [the midwife] and she'll see what to advise you so that you can restore the peace at home. [After] the disagreements that existed at home, respect has settled into the home now; that too made me happy."

Focus group participants reported that if a woman had financial difficulties that prevented her from travelling to the health center to attend a G-ANC meeting, or from paying the ANC fees, she could inform women they were paired with in the same G-ANC group, and they did what they could to help:

"There are some people, because of the [costs of] travel [to the health center], they will say that the place is too far, they will say this and that. If you can't find [money to pay for transport] and you call your second [the other woman in your pair], if she really has it [transportation money] or even if she doesn't have it, she will try to help you find a solution…"

When a woman was absent, the provider and the woman she was paired with contacted her and tried to assist with the problems she was facing. This might include home visits by the other woman they were paired with, advice, provision of financial assistance, waiving of consultation fees by the midwife, or conflict mediation with her husband.

**Survey data on acceptability, self-efficacy, and comparison to qualitative findings**

Nearly all (96%; 134/140) of the women in the endline household survey who reported having participated in G-ANC said they would choose G-ANC again if they had another pregnancy. Women noted that they preferred G-ANC to individual ANC because they enjoyed being part of a group (74%), felt they received more comprehensive care (63%), and appreciated developing a relationship with one provider (49%) (Table 2).

More women who participated in G-ANC than those who had individual ANC reported that it was "mostly true" or "completely true" that they were given an opportunity to ask questions about family planning (p-value < 0.01) and had all their questions answered in a way they could understand (p-value = 0.02; Table 3). Despite focus group participants' higher satisfaction with the education and advice from their providers compared with those who attended individual ANC, only 30.7% of endline survey respondents reported that it was "somewhat true" or "mostly true" that all their questions were

**Table 2. Pros and cons of group antenatal care reported during endline household survey.**

| Item | Reason | Select-All Responses (% of Respondents; totals may exceed 100%) |
|---|---|---|
| Aspects of Group ANC that women liked, among those who would choose it again (N = 134) | Enjoyed being part of a group | 99 (74%) |
| | Felt care was more comprehensive | 84 (63%) |
| | Liked relationship with same provider | 65 (49%) |
| | Received reminders to attend | 25 (19%) |
| | Other not specified | 6 (4%) |
| Amongst women who would not choose G-ANC again in a future pregnancy, why not? (N = 6) | Too many visits | 2 (33%) |
| | Too much time | 2 (33%) |
| | Did not like being in the group | 1 (17%) |
| | Not enough privacy | 1 (17%) |

**Table 3. Women's agreement with self-efficacy statements according to reported participation in G-ANC.**

| | Individual ANC | G-ANC | p-value |
|---|---|---|---|
| **Total** | **2376** | **140** | |
| In interactions with health providers, I was given the opportunity to ask questions about family planning methods | 232 | 45 | 0.007 |
| | 20.5% (17.0-24.0) | 30.7% (23.0-38.5) | |
| In interactions with health care providers all of my questions about family planning were answered in a way I understood | 233 | 42 | 0.026 |
| | 20.7% (17.3-24.1) | 28.7% (21.1-36.2) | |

answered in G-ANC, and only 28.7% said their questions were answered in a way they could understand. Focus group participants nevertheless described improvements in their understanding and comfort level with key behaviors.

Endline survey responses on pregnant women's interactions with health care providers on family planning, according to type of ANC service delivery received (individual ANC or group ANC). These responses were collected on a five-point scale and were dichotomized with not at all true through somewhat true indicating disagreement (0) vs. mostly or completely true indicating agreement [1].

## Content domains of self-efficacy

We identified four content domains of self-efficacy that G-ANC processes targeted: 1) timely care-seeking, 2) pregnancy management, 3) pregnancy self-care, and 4) facility delivery (Table 4).

Higher self-efficacy for timely care-seeking improved women's autonomy to seek medical attention quickly, even if their husband's support was needed to access care:

"The confidence that there is in it for me, [the nurse who conducts the deliveries] told us that [a pregnant patient] had the heat [fever] and came and that [the nurse] had prescribed her a drug, but once [the pregnant woman] arrived at the pharmacy, she did not buy the medicine and instead [bought] Paracetamol, and when it was time to give birth, she took the child like that, like that in a bad state, and that did the child a lot of harm. It scared me. So, when I had heat [fever], I was quick to come and when she [the nurse] prescribed things for me, I quickly bought them. If I hadn't heard this [story], I would have stayed home and started collecting and drinking some traditional medicine."

"...it's me who feels the pain who must say 'I must go to the hospital' before my husband tells me to go to the hospital, and 'give me some money so that I can go to the hospital.'"

**Table 4. Four content domains of self-efficacy identified by pregnant women in focus groups.**

| Self-Efficacy Domain | Definition (Confidence that one can…) | Illustrative Quotes |
|---|---|---|
| **1. Timely Care-Seeking** | take steps to rapidly access care when danger signs appear; overcome stressful situations such as physical pain; advocate for oneself with partner and/or other members of the household | "It's me who feels the pain who must say, 'I must go to the hospital' before my husband tells me to go to the hospital and 'give me some money so that I can go to the hospital.' Because even if you tell them that, they don't believe it, it's you who will get up (make the decision) to go to the hospital."<br>"We have to quickly look for the motorcycle taxi that will take us [to the health center] because there are some people who it's only when the baby thing starts to sting them that they start looking for a motorcycle and this ends up leading to other things [complications] and the child may no longer be found [the child may be stillborn], or the child's mother may no longer be found [the mother may die]."<br>"The confidence that there is in it for me, [the nurse who conducts the deliveries] told us that [a pregnant patient] had the heat [fever] and came and that [the nurse] had prescribed her a drug, but once [the pregnant woman] arrived at the pharmacy, she did not buy the medicine and instead [bought] Paracetamol, and when it was time to give birth, she took the child like that, like that in a bad state, and that did the child a lot of harm. It scared me. So, when I had heat [fever], I was quick to come and when she [the nurse] prescribed things for me, I quickly bought them. If I hadn't heard this [story], I would have stayed home and started collecting and drinking some traditional medicine." |
| **2. Pregnancy Management** | monitor, interpret, and take appropriate action in response to pregnancy symptoms | "It [G-ANC] has [given me] confidence because when we come, if something surprises you by chance, you can explain, if you don't want to say publicly, you will tell the person who looked at [examined] you and such as you will do, she will explain it to you."<br>"When a woman is pregnant and is taking leaf water from home, we know what leaf water from home does to us. Some of us know what it's about, it can cause you to give birth like this, like this [in a poor state of health] because the person who wants to fetch you the [medicinal] plant water doesn't know your organism, so if he fetches you the plant water, it can create something else [complications, adverse effects on the health of the pregnant woman and/or the fetus]. We're going to say that we're wanting to put the baby to sleep and, on the contrary, waking him up [aggravating an illness by wanting to cure it, or creating a new health problem by wanting to solve another]. So, I see that if we come directly to the caregiver's home [health center], it's better." |
| **3. Pregnancy Self-Care** | Take necessary health precautions and follow recommendations to achieve a positive pregnancy outcome, such as taking IPTp | "In the group they taught us that we have to take care of ourselves and sleep under a mosquito net regularly."<br>"If I'm prescribed medicine… and my husband doesn't have the money to buy it in time, if I look in myself, I can take money to buy, because I have to remember my well-being and that of the child."<br>"This last time when I gave birth, I didn't miss any of the [doses of] medications [to prevent malaria]."<br>"Before, I had difficulty taking it, and when we came here for the meetings, I reported this to the midwife, and the midwife said that when I come to take it another time, to… take it... with a lot of water, and that if I took enough water it wouldn't bother me. And it's true, when I took it with enough water, about half a liter, it didn't bother me. I haven't vomited anymore since that time."<br>"[G-ANC] made it so that when I wanted to give birth, it wasn't difficult at all, because when I came to the [group] meetings and I had check-ups, I would buy and use the medicines as prescribed. As it was happening, I had it explained step by step so that I wouldn't have any more problems." |

*(Continued)*

| Self-Efficacy Domain | Definition (Confidence that one can…) | Illustrative Quotes |
|---|---|---|
| **4. Facility Delivery** | insist on a facility delivery and reach facility when going into labor, self-advocacy | "If you don't do anything yourself to get money, you won't be able to get out of it, but if you're responsible for yourself, when it starts, you'll take responsibility and tell yourself that you have to go to the hospital to have peace. That's how I did it. When you're at your husband's [mercy], some men neglect people."<br>"When the thing of the baby is going to start to sting [when contractions start], even if your husband doesn't have any money, you'll quickly get a motorcycle taxi to get to [the health center]… sometimes, because of money, things that shouldn't happen do happen. So, we've been told that we have to buy a box and we'll be contributing a little, a little money in there."<br>"We can make [a pregnant woman] leaf water [herbal decoctions] at home, but even if we do that at home, we have to come to the caregiver's house [the health center] because the caregiver is our second god in the world here." |

In the domain of pregnancy management, G-ANC increased participants' confidence to monitor, interpret, and take appropriate action in response to pregnancy symptoms, and provided pregnant women assurance that they could handle pregnancy, childbirth, and newborn care:

"Yes, I'm more confident that if I get pregnant, I could take care of the pregnancy right up to the time of delivery."

G-ANC participants discussed improved self-efficacy to take IPTp, an aspect of pregnancy self-care. Many women reported that during previous pregnancies, they did not take prescribed medications and supplements. Women reported feeling more confident about taking iron supplements as well as IPTp, and making greater efforts to take medications while enrolled in G-ANC:

"Yes, I take all [of the drugs offered at the health center]… At the beginning, I had difficulty taking the drugs, but as I started attending the meetings, the teachings, I started making the effort to take the drugs."

In the fourth content domain, women described a greater ability to reach the health facility in time to give birth in a facility:

"When the thing of the baby is going to start to sting [when contractions start], even if your husband doesn't have any money, you'll quickly get a motorcycle taxi to get to [the health center]… sometimes, because of money, things that shouldn't happen do happen. So, we've been told that we have to buy a box and we'll be contributing a little, a little money in there. After that, we have to quickly look for the motorcycle taxi that will take us [to the health center] because there are some people who it's only when the baby thing starts to sting them that they start looking for a motorcycle and this ends up leading to other things [complications] and the child may no longer be found [the child may be stillborn], or the child's mother may no longer be found [the mother may die]."

**Sources of efficacy expectations: performance accomplishments.** G-ANC generated performance accomplishments when women experienced success at measuring and recording weight and vital signs, taking IPTp during meetings, and trying different behaviors at home that they learned in the group. For new behaviors performed outside of meetings, G-ANC participants reported back to the group on the recommended behaviors they had practiced on their own at subsequent meetings.

G-ANC participants reported pride in acquiring skills that made them feel more confident and valuable to the community. For example, women expressed self-satisfaction regarding how they had learned to take each other's vital signs:

"What I have gained in it, I myself know how to take the temperature of my neighbor, I myself know how to measure weight, how to take blood pressure, I know all this, whereas I didn't know how to do anything before."

**Sources of efficacy expectations: vicarious experience.** Vicarious experience during G-ANC came from watching health aides and other women in the group take one another's vital signs, observing others take IPTp, role-playing, and sharing personal experiences:

"Sometimes they put on theater [role play], and we watch it as an example played out between the midwife and the *Yovonon* [nickname for a health aide who takes care of the delivery]."

During time for sharing, women spoke of acquaintances who had poor pregnancy outcomes because they failed to address danger signs; participants offered advice from their previous pregnancies, and women who returned to the group after childbirth explained how lessons from G-ANC had helped them.

**Sources of efficacy expectations: verbal persuasion.** Because frequent ANC attendance throughout pregnancy is considered a key behavior to achieve proper pregnancy management, helping women overcome barriers to attendance is crucial. Encouragement by the women they were paired with in G-ANC meetings motivated participants to attend more ANC:

"When you arrive at a [G-ANC] meeting, and you don't see your second [the other woman in your pair] with whom you meet, when you start phoning them and you call them until… They always end up making themselves available, and we all come together."

Participants explained that they also received advice from trusted providers who facilitated G-ANC:

"When I took [IPTp] for the first time and I vomited; [the midwife] told me, if I have to take it another time, to not eat anything before taking it, and I followed that advice."

Focus group participants consistently remarked on how much more time and care their providers took to give them detailed advice and explanations during G-ANC meetings, as compared to individual ANC. For instance, some women reported that the drug given for IPTp was bitter, and pills had been difficult to swallow or caused undesirable effects. After confiding in their healthcare providers during G-ANC, they received coaching that helped minimize these effects and facilitated behavioral adoption.

**Sources of efficacy expectations: emotional and physiological states.** Focus groups did not describe stress or anxiety around taking IPTp and practicing other behaviors. They did however describe a positive emotional state induced by G-ANC that may have helped them to assimilate new information presented during meetings [14,20].

Endline survey analysis found no significant difference between individual ANC and G-ANC participants when it came to health behaviors that they believed they had the self-efficacy to perform, such as successfully asking for IPTp (p-value 0.18) and taking IPTp at least three times during pregnancy (90 vs. 93.3%, p-value 0.29, Table 5). Belief in the outcome efficacy of taking IPTp was high whether or not women had participated in G-ANC (p-value 0.64, Table 5), as were other outcome efficacy beliefs. The only significant difference found was in recognizing the importance of taking IPTp while also sleeping under a bed net consistently (100% of G-ANC participants vs. 95.6% of individual ANC participants [p-value 0.01, Table 5]).

**Table 5. Perceived self-efficacy and outcome efficacy.**

| Action | Individual ANC | Group ANC | p-value |
|---|---|---|---|
| Go for antenatal care as soon as I think I might be pregnant | 80.9% (78.2, 83.7) | 81.2% (74.7, 87.7) | 0.93 |
| Convince my spouse/partner to accompany me to the health facility for antenatal care | 75.6% (72.4, 78.9) | 76.7% (70.6, 82.9) | 0.77 |
| Take the medicine to prevent malaria at least three times during pregnancy | 90% (87.3, 92.7) | 93.3% (88.9, 97.7) | 0.29 |
| Request for the medicine that helps to prevent malaria when I go for antenatal care | 83.9% (81.4, 86.5) | 89.6% (83.4, 95.7) | 0.18 |
| The medicine given to pregnant women to prevent malaria works well to keep the mother healthy. | 98.5% (97.8, 99.2) | 96.5% (93.1, 99.9) | 0.09 |
| The medicine given to pregnant women to prevent malaria works well to make sure her baby is healthy when it is born | 98.1% (97.2, 99.1) | 98.7% (96.8, 100) | 0.64 |
| Pregnant women should still take the medicine that is meant to keep them from getting malaria even if they sleep under nets every night | 95.6% (93.9, 97.3)* | 100% (100, 100)* | 0.01* |

* Indicates a statistically significant difference using a 95% confidence interval.

Self-efficacy beliefs about IPTp were higher than self-efficacy beliefs about accessing ANC in general, by 10 percentage points. This emerged in the household surveys when women were asked if they believed they could go for ANC as soon as they thought they might be pregnant (81.2%, 95% CI 74.7-87.7) and if they believed they could convince their partners to accompany them to the health facility for ANC (76.1%, 70.6-82.9), as compared to results for their belief in the ability to request IPTp at ANC (86.8%, 81.4-95.7), and in the ability to take all three doses of IPTp (91.7%, 87.3-97.7).

## Discussion

Grenier et al. theorized that pride in new knowledge, skills, and the ability to act, along with increased social capital and support gained from G-ANC meetings, could lead to improved self-care and care-seeking practices [3]. Opportunities in G-ANC to take blood pressure, observe other participants, and receive verbal persuasion, especially from the women they were paired with, may have provided sources of self-efficacy that empowered women to overcome barriers and perform recommended health behaviors. The measurable impact of G-ANC on health literacy and operationalization of health education has been borne out previously in other studies in sub-Saharan Africa [12,21]. With few exceptions [4,5], though, G-ANC has not shown measurable improvements in the Pregnancy-Related Empowerment Scale (PRES) elsewhere in sub-Saharan Africa [5–7].

A realist evaluation on the mechanisms of effect for group models of antenatal care produced a Pregnancy Circles Logic Model with increased self-efficacy occupying a prominent place in the model, and identified it as one of six intermediate outcomes in the model [22] rather than as a mechanism along the causal pathway in the model of Grenier et al. [3]. The intermediate outcomes result in turn from six mechanisms of change including group cohesiveness, health literacy, and psychological empowerment [22]. A scoping review on G-ANC for adolescents identified empowerment as a key theme, and self-efficacy as one contributor to empowerment [23]. This scoping review in turn identified two studies that highlighted self-efficacy as a mechanism for the effects of G-ANC in adolescents [24,25]. In many other studies including in Nigeria [26] and Ghana [27], self-efficacy is not reported in the factors affecting effectiveness [27], or is not present in the theory of change as either a mechanism or an outcome [26].

While qualitative methods have provided evidence that women in G-ANC have a better experience of care and described feeling more capable of performing healthy behaviors, quantitative methods — generally employing Likert scales — have not shown a significant improvement in G-ANC participants' self-efficacy [4–7]. The same held true for this nested study of self-efficacy in Benin. This recurring lack of significant increase in self-efficacy is counter-intuitive, given that the processes that take place in structured G-ANC meetings align with Bandura's sources of self-efficacy.

In Benin, midwives, nurses, and health aides who facilitated G-ANC meetings helped build participants' self-efficacy during G-ANC meetings through hands-on practice, role playing, social support, and encouragement. They also induced a positive emotional state that may have helped reinforce new knowledge acquired through G-ANC meetings and encouraged more ANC contacts by G-ANC participants. In focus group discussions, women felt they had not only acquired knowledge, but also improved their capacity to overcome certain barriers to care. For instance, collective problem-solving and social learning similar to that described by Bandura [14,28–31] helped women strengthen their ability to convince partners to let them attend regular ANC visits and seek care for pregnancy danger signs.

G-ANC participants were able to find help if they needed it by confiding in providers and peers. Social bonds formed between G-ANC participants and group facilitators emerged as a key benefit of G-ANC as a service delivery model. Consequently, G-ANC participants were more likely to seek providers' help in navigating and managing their pregnancies. Social support from G-ANC helped them to find ways of accessing care at the health facility. Participants described enhanced self-advocacy and exercising autonomy to overcome barriers to preventive care and care-seeking (e.g., saving their own money for facility-based childbirth). The knowledge and confidence gained were essential given the cultural context in which a woman often relies on her husband's knowledge and opinions regarding health care. Still, an endline survey showed that G-ANC participants remained less confident in their ability to convince partners to attend ANC with them than in their ability to perform other pregnancy-related behaviors that they could do on their own.

Focus group participants in Ghana reported feeling that G-ANC facilitators answered their questions completely [21]; although some focus group participants in Benin highlighted getting more comprehensive answers to their questions during G-ANC meetings, this was not the case for respondents to the endline household survey, of whom only one-third said all of their family planning questions were answered. Relative to the Bandura framework, one can infer from the indication that many women did not get all their questions answered that verbal persuasion was likely not used optimally as a source of self-efficacy. It is important to note, however, that the only standardized malaria behavior survey item used in the endline survey in Benin asked about interactions with providers on their family planning questions. The survey did not inquire about the completeness of answers to women's questions on other topics, e.g., pregnancy symptoms or malaria prevention. Participants in focus group discussions in Benin nevertheless said they learned and applied many new practices in G-ANC, which they had not known or done during past pregnancies.

Similar to past studies from 11 low- and middle-income countries indicating G-ANC provides a better experience of care overall [32,33], participants in Benin considered the care they received more comprehensive than in individual ANC. Women reported feeling that they developed both more detailed knowledge and greater confidence to perform health behaviors in four domains: timely care-seeking; pregnancy management; pregnancy self-care; and facility delivery.

Quantitatively, while population-level coverage of G-ANC in Atlantique Department was low (14% in the intervention arm compared to a target of 50% of all pregnant women attending their first ANC consultation) [15], the endline household survey revealed that women who attended G-ANC had a more positive experience relative to their prior pregnancies, and 96% would prefer G-ANC over individual ANC in a future pregnancy [15]. Studies in Bangladesh, Burkina Faso, Egypt, Iran, Kenya, Malawi, Nepal, Nigeria, Rwanda, Senegal, and Tanzania had similar findings about women's preference for G-ANC over individual ANC, with a large majority indicating that they were more satisfied or had a better experience of care overall in G-ANC [7,9,32,34,35].

Efficacy-related items included in the parent study's endline survey showed that over 98% of women in both study arms shared high outcome expectations that IPTp worked well to make sure one's baby would be born healthy. Likewise, women's self-efficacy to take IPTp at least three times during pregnancy was 90% even in the control arm; it is relevant to note that women made affirmative self-efficacy statements and did not express fear or anxiety as the main barriers to this behavior in focus groups. Among pregnant women surveyed who participated in the G-ANC intervention, the parent study found a significant association with improved uptake of both ANC4 and taking three or more doses of IPTp. G-ANC in Benin was linked to a higher mean number of IPTp doses taken, compared with prior research in Nigeria and Kenya [10].

Endline survey results measured slightly higher outcome expectancy among G-ANC participants that taking IPTp in addition to sleeping under a mosquito net could prevent malaria. This finding points to knowledge gains; on the other hand, there was no significant difference in self-efficacy expectations among G-ANC and individual ANC participants when it came to convincing their partners to accompany them to ANC, asking for IPTp at the health facility, or taking at least three doses of IPTp while pregnant.

Top benefits of participating in G-ANC identified by Beninese women in response to a multiple-choice question in the endline survey included group cohesion and solidarity, a positive emotional state while attending meetings, provider support, mutual aid between group members, and exposure to beneficial behaviors like taking IPTp. Evidence from the parent study and this nested study suggest the factors that led G-ANC participants to take more IPTp during their pregnancies likely related to their accessing more ANC in general [15]. Social learning (i.e., learning behaviors through observation and modeling) entails a series of parallel processes [13,14,28–30], and the positive emotional state that focus group participants reported during group meetings may have given women greater pleasure-based motivation to attend G-ANC [20]. Persuasion by providers and peers to return for additional G-ANC meetings appears to be another component that led women to take more doses of IPTp.

## Study limitations

The population-based household survey included only a small number of the women who opted to participate in G-ANC when it was offered during their first ANC visit. Therefore, we are unable to perform pre-post comparisons of self-efficacy of women who attended G-ANC, and we cannot assess whether G-ANC participants might have had higher self-efficacy at baseline, compared to the non-participants. The quantitative endline survey found evidence that most eligible women were not given the option to participate in G-ANC by their provider, thereby reducing the sample size for detecting a significant effect of G-ANC on self-efficacy. Furthermore, social desirability bias may affect some responses, such as the elevated proportion of women in both study arms (over 98%) stating that IPTp worked well to make sure one's baby would be born healthy.

The qualitative interviews and focus groups were not designed specifically to investigate the effect of G-ANC on women's self-efficacy; the line of inquiry on self-efficacy was a secondary qualitative analysis.

Lastly, Bandura's definition of self-efficacy focuses on word choice to distinguish between perceived capability and intention. Bandura emphasizes the difference between "will" and "can," with "will" signaling intent and not self-efficacy, while "can" indicates perceived capability. The Fongbe verb *sixu* does not map directly onto the French verb *pouvoir* or the English verb *to be able*. Therefore, self-efficacy statements identified in the focus group transcripts as part of this secondary analysis may not accurately reflect Bandura's English-language criteria for such statements.

## Conclusions

In this first formal study on G-ANC in Benin, we found G-ANC to be acceptable to and preferred by women in Atlantique Department who participated. This coincides with findings from other settings regarding acceptability and higher experience of care overall in G-ANC [7]. This study also sought to identify ways that G-ANC might foster changes in women's propensity to adopt or maintain preventive and care-seeking behaviors; prior studies of empowerment and self-efficacy have shown that G-ANC improves health literacy [21] and empowers women in some settings [5].

The endline survey analysis performed for this nested study found no significant difference between individual ANC and G-ANC participants when it came to malaria prevention and other health behaviors that they believed they had the self-efficacy to perform. Self-efficacy to take IPTp was already high at baseline and thus needed less reinforcement, as were outcome efficacy beliefs. The one significant difference in self-efficacy among G-ANC participants was in recognizing the importance of taking IPTp while also sleeping under a bed net consistently (100% of G-ANC participants vs. 95.6% of individual ANC participants [p-value 0.01, Table 5]).

However, according to qualitative findings some of the processes inherent to G-ANC seem to have strengthened self-efficacy in other domains. We identified four domains of self-efficacy strengthened by G-ANC participation: timely care-seeking; pregnancy management; self-care; and facility delivery. G-ANC processes that empowered women to overcome barriers to ANC attendance and provided social support and encouragement to return to ANC (i.e., higher self-efficacy to seek care) may have facilitated their taking more doses of IPTp. While participants in G-ANC reported more opportunities to ask questions than those in individual ANC, future G-ANC training should emphasize that facilitators need to allow more time for questions in group and individual exam segments, answer in a way that women can understand, and focus on the parallel processes of G-ANC meetings that foster self-efficacy.

## Supporting information

**S1 Appendix. Group antenatal care processes and their intended effect on self-efficacy.**
(DOCX)

**S2 Appendix. Comparison of individual antenatal Care (ANC) and group antenatal care (G-ANC) as implemented in Benin.**
(DOCX)

**S3 Appendix. Health facility characteristics considered for purposive sampling.**
(DOCX)

**S4 Appendix. Tool 05: Focus group with pregnant women and women who have recently given birth.**
(DOCX)

**S1 File. Inclusivity in Global Research.**
(DOCX)

## Acknowledgments

The authors would like to acknowledge the health care workers, supervisors, and pregnant women who participated in this study, as well as 18 research assistants, the Benin Ministry of Health, Health Zone authorities (Abomey-Sô-ava, OKT, Allada-Toffo-Ze), and Shelby Cash of CDC.

## Author contributions

**Conceptualization:** Julie Niemczura Sutton, Courtney Emerson, Stephanie Suhowatsky, Julie Buekens, Katherine Wolf, Cyriaque D. Affoukou, Julie R. Gutman, Peter John Winch.

**Data curation:** Julie Niemczura Sutton, Fifamè Aubierge Eudoxie Kpatinvoh, Esther Firmine Cadja Dodo, Erin Go, Peter John Winch.

**Formal analysis:** Julie Niemczura Sutton, Fifamè Aubierge Eudoxie Kpatinvoh, Esther Firmine Cadja Dodo, Erin Go, Courtney Emerson, Peter John Winch.

**Funding acquisition:** Julie R. Gutman.

**Investigation:** Fifamè Aubierge Eudoxie Kpatinvoh, Esther Firmine Cadja Dodo, Julie R. Gutman.

**Methodology:** Courtney Emerson, Julie R. Gutman, Peter John Winch.

**Project administration:** Julie Niemczura Sutton, Catherine Dentinger, Julie Buekens, Manzidatou Alao, Marie Adeyemi Idohou, Faustin Onikpo.

**Supervision:** Julie Niemczura Sutton, Catherine Dentinger, Manzidatou Alao, Marie Adeyemi Idohou, Faustin Onikpo, Cyriaque D. Affoukou, Aurore Ogouyèmi-Hounto, Julie R. Gutman, Peter John Winch.

**Validation:** Fifamè Aubierge Eudoxie Kpatinvoh, Esther Firmine Cadja Dodo, Faustin Onikpo, Peter John Winch.

**Writing – original draft:** Julie Niemczura Sutton, Erin Go.

**Writing – review & editing:** Fifamè Aubierge Eudoxie Kpatinvoh, Courtney Emerson, Catherine Dentinger, Stephanie Suhowatsky, Julie Buekens, Katherine Wolf, Cyriaque D. Affoukou, Aurore Ogouyèmi-Hounto, Julie R. Gutman, Peter John Winch.

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
