## [Decision Letter · Decision Letter 0]

18 Jul 2025

PGPH-D-25-01395

“Each moon we come to weigh the pregnancy.” Self-efficacy in group antenatal care in Benin

Dear Dr. Winch,

Thank you for submitting your manuscript to PLOS Global Public Health. After careful consideration, we feel that it has merit but does not fully meet PLOS Global Public Health’s publication criteria as it currently stands. Therefore, we invite you to submit a revised version of the manuscript that addresses the points raised during the review process.

A rebuttal letter that responds to each point raised by the editor and reviewer(s). You should upload this letter as a separate file labeled ’Response to Reviewers’.A marked-up copy of your manuscript that highlights changes made to the original version. You should upload this as a separate file labeled ’Revised Manuscript with Track Changes’.An unmarked version of your revised paper without tracked changes. You should upload this as a separate file labeled ’Manuscript’.

We look forward to receiving your revised manuscript.

Kind regards,

Nicola Hawley

Academic Editor

Journal Requirements:

Additional Editor Comments (if provided):

Reviewers’ comments:

Reviewer’s Responses to Questions

**Comments to the Author**

1. Does this manuscript meet PLOS Global Public Health’s publication criteria? Is the manuscript technically sound, and do the data support the conclusions? The manuscript must describe methodologically and ethically rigorous research with conclusions that are appropriately drawn based on the data presented.

Reviewer #1: Partly

Reviewer #2: Yes

2. Has the statistical analysis been performed appropriately and rigorously?

Reviewer #1: I don’t know

Reviewer #2: Yes

3. Have the authors made all data underlying the findings in their manuscript fully available (please refer to the Data Availability Statement at the start of the manuscript PDF file)?

Reviewer #1: No

Reviewer #2: No

4. Is the manuscript presented in an intelligible fashion and written in standard English?

Reviewer #1: Yes

Reviewer #2: Yes

5. Review Comments to the Author

Reviewer #1: Thank you for the opportunity to review this manuscript. I found it very interesting and a great value add to the literature! Regarding the questions above, the statistical tests used were not specified, which is why I answered "I don’t know" to question 2. The authors state that their data availability plan is still under review, which is why I answered "no" to question 3. While there is a lot of great data and discussion in this manuscript, I think there needs to be substantial reorganization to ensure clear understanding of the aims, the methods for each aim, and the findings within each aim. I have provided specific comments for each section below and in the document attached:

Introduction

• To improve the flow, I recommend discussing outcomes (lines 59-64) and then introducing the theory of change (lines 57-59) as a mechanism for why we see these outcomes. For readers who are unfamiliar with the theory of change proposed by Grenier et al., I think it would be beneficial to elaborate on how G-ANC increases self-efficacy in 1-2 sentences. This then ties into you introducing/defining self-efficacy in lines 65-74.

• I would like to see more discussion in the paragraph about G-ANC processes (75-79). You correctly specify that “theoretically” the processes are taking place, and you lay them out well in your Table 1 and Figure 1. I feel like there is a real opportunity here to highlight the gap in the literature that you are filling with this manuscript. Not many other G-ANC studies, particularly in LMICs measure self-efficacy, but given its crucial role in the theory of change, this paper is a great value add to the literature.

• In lines 80-85, I recommend parsing out the aims of the parent study and the aims of this paper, specifically. In their current form, the wording of these three do not directly match the results.

Methods

• For clarity, I would discuss the quant and qual aspects of the paper in the same order. For example, qual aims then quant aims, qual methods then quant methods, qual analysis, then quant analysis, etc. – there seems to be some switching back and forth.

• You mention that the focus groups were conducted in Fongbe and transcribed to French (lines 120-121) – were they then translated to English for analysis? It seems like the deductive analysis was using English word codes – was the coding done in English or French?

• In your study limitations section, you state that interviews and focus groups were not designed to investigate the effect of G-ANC on women’s self-efficacy, so I would consider re-framing this as a secondary qualitative analysis.

• In lines 133-134, you briefly discuss the software used for quantitative data collection and analysis, but you do not discuss what analyses were performed. For example, what statistical tests were used to get the p-values in Table 3?

Results

• Lines 148-163 feel a bit out of place to me. You reference Table 2 at the end of line 150, but Table 2 has no mention of social support, stressful situations, or self-advocacy, which is what the preceding sentences are referring to. Are these supposed to be overall insights into women’s experiences (related to Aim 1)? If so, incorporation of themes would be helpful. I am not quite getting the link between what is discussed here and self-efficacy, if it is supposed to be connected.

• The content domains of self-efficacy and the sources of efficacy expectations were very clear.

• The connection between the quant survey findings and the qualitative findings is a bit unclear for some measures. It may be that there needs to be more information in the introduction section about how certain survey indicators are related to self-efficacy. How do the outcomes in line 240-244 relate to self-efficacy? Additionally, you present data on the statement: “I was given the opportunity to ask questions about family planning methods.” To me, this more so speaks to provider quality and fidelity to the curriculum and less about self-efficacy. I was also curious why there was a quantitative focus on family planning when this did not come up at all in any of the qualitative data?

• I do not see Table 4 referenced in the text. The data presented in Table 4 is very interesting, but I am not quite getting the connection to self-efficacy. Perhaps it is related to one of the other aims? Overall, I would like to see the aims, the methods for each aim, and the results for each aim more clearly outlined.

Discussion

• The discussion could also be reorganized to reflect the findings of each aim. I recommend starting out with exactly what you found and then related it back to the theory of change and other literature. You were going back and forth between the qual and quant findings, which was a bit confusing.

• Great job connecting to other studies in LMICs! However, I think you could be a bit more detailed. For example, in lines 310-312, you state “Studies in Nepal, Senegal, Bangladesh, and Rwanda had similar findings.” Similar findings in that women had more positive experiences or that they would prefer G-ANC over individual ANC. Each study words their findings / measure these concepts in different ways, so you may want to explicitly discuss what each found.

Conclusion

• The first sentence (line 352) makes me think assess acceptability was supposed to be a main aim of this paper. Maybe that is a part of you aim 1? If this is the case, maybe aim 1 should be reworded and the title changed to reflect the focus beyond self-efficacy.

Reviewer #2: Title: The selected quote in the title does not match the data presented, so it does not really reflect the results or take home messages from the paper

Introduction:

The linking of Bandura and Grenier theory of change is innovative.

Clarify last paragraph because as written implies this is a qualitative study, but qualitative and quantitative results are presented [A nested qualitative study was conducted alongside a cluster randomized controlled trial assessing the impact of G-ANC on IPTp and ANC attendance in Atlantique Department, Benin (Gutman, 2024)]

A little background on the behaviors that came into focus for the content areas of self-efficacy and how these link to improved pregnancy outcomes is needed because at times malaria (SP for IPTp) seems like the main concern, but then family planning, facility delivery, emergency seeking and attendance are included. Clarity about self-efficacy and linkages to these various behaviors is needed to set the stage. Not a lot is needed, but enough to understand how these relate to improved pregnancy outcomes in general and specifically for Benin (especially since malaria-related data as presented seem fairly good, so from an intervention perspective, a very large sample would be needed to know the effect on this particular outcome).

Methods:

Is this specific mixed methods design convergent, i.e., intending to validate findings from one method with results from the other? To gain a more holistic view of the self-efficacy? I would not call this a qualitative study; it seems to be mixed methods

Given data are presented from both G-ANC and Individual, basic details about the RCT are needed [primary outcome of RCT?].

A basic description of individual ANC is needed to compare to G-ANC

More details for the intervention would be helpful to know (# of sessions, schedule of sessions, types of facilitator, curriculum same/different than recommended by MoH?). Since the focus is on specific behaviors how different or similar was the specific content compared to usual ANC? Was the time allocated to malaria and prevention the same as usual ANC or did those in G-ANC get more time for education?

If this is a CenteringPregnancy adaptation, what were the core components and was fidelity also considered? If so, how does it relate to interpretations (or limitations)?

What were the questions from FGDs (interview guide)?

A simple table or supplemental material with the basics about the six selected, indicating the purposefulness of the purposeful sampling, would be helpful [how these relate to the 20 in the larger would be good to know too]

Use of framework analysis is appropriate given the theory-driven a priori categories of Bandura’s self-efficacy, but a bit more detail about how this was done procedurally is needed in methods and data management and analysis paragraphs for FGDs. For example, were all transcripts read in full, or were "keyword" searches done, or a combination of these? What were all the word choices? How was the matrix setup?

Results:

The quotes on Table 2 are great. However, the quote in the text, "If you have a problem at home, you can tell [the midwife] and she’ll see what to advise you so 152 that you can restore the peace at home. [After] the disagreements that existed at home, 153 respect has settled into the home now; that too made me happy.” - does not seem to provide evidence that the woman was feeling more confident about advocating for herself. The advice from the midwife might have just been about conflict resolution or some other way of dealing with disagreements, but not necessarily an indicator of being able to advocate. If that is not the purpose then transition sentences with rationale to understand the selection of a quote would help.

Table 3: Just confirming - the sample sizes for Individual is 2376 and G-ANC is 140 - the differences shown are statistically significant (and wider confidence intervals for G-ANC indicate this). Even though there seems to be enough power since it is significant, contextualization of the methods would help to make sense, since it seems that all those in G-ANC would have taken the survey too, not just a subsample from RCT. Unclear how the data in this paper relates to the larger study in which these are embedded. It may be that there is confusion because the household surveys were not part of the RCT. How many in the survey did not attend 2 or more ANC visits since this was an (appropriate) inclusion criterion for the 140? Also, assuming these were Chi-square and Fisher’s exact tests, but I did not see that stated anywhere on the table or in the text.

Line 235 under Survey Data on Self-Efficacy, Outcome Efficacy, and Comparison to Qualitative Findings - Did it attend at least one G-ANC or two, as stated earlier in the manuscript?

Lines 243-44 imply continuity of care - were the same providers assigned to each group?

Discussion:

Line 280: I think more than 16 LMICs have tried G-ANC - update to include articles from other LMICs can be integrated to support these data and the discussion - e.g., Ghana, Burkina Faso, Malawi, and Haiti and I found one from Tanzania that also includes data about IPTp (although it appears to have not been peer reviewed yet)

The discussion needs to go a bit deeper to address the why behind the communication gaps = if was a four visit model then could be that four visits are not enough and supports WHO’s recommendation for eight visits, but number of visits is not known.

What does the population in Benin look like from a reproductive point of view to know how representative the 140 people in the study were (since nearly all married, 26.7 years old, and mean of 2.8 children)?

Limitations:

How was social desirability considered, particularly in the FGDs?

General:

The word convenes is used a couple of times and is an odd word choice for health service settings

The use of "participatory education and antenatal care" implies that the health promotion aspects of G-ANC are separate from the care, instead of part of the care. Consider rephrasing

The abstract states that, "A qualitative study was embedded in a trial assessing the impact of G-ANC on ANC retention and intermittent preventive treatment of malaria in pregnancy (IPTp) uptake to assess women’s experience of GANC, and ways participation could foster self-efficacy" but this paper is bigger than this and offers more than insights that those of IPTp (i.e., emergency care-seeking, pregnancy management, pregnancy self-care, and facility birth), so may be too narrow of a goal given the data presented. It is hard to tell when data from the parent study vs this targeted study. It may be that the only behaviors measured objectively were IPTp and attendance, but that this paper explored the mechanisms identified in Grenier related to support and building of confidence, i.e., self-efficacy (qualitative and some other quantitative), so clarity is needed.

6. PLOS authors have the option to publish the peer review history of their article (what does this mean?). If published, this will include your full peer review and any attached files.

**Do you want your identity to be public for this peer review?** For information about this choice, including consent withdrawal, please see our Privacy Policy.

Reviewer #1: No

Reviewer #2: No

---

## [Decision Letter · Decision Letter 1]

24 Feb 2026

PGPH-D-25-01395R1

“Each moon we come to weigh the pregnancy:” Exploring the experience of group antenatal care processes in Benin and their contributions to self-efficacy

Dear Dr. Winch,

Thank you for submitting your manuscript to PLOS Global Public Health. After careful consideration, we feel that it has merit but does not fully meet PLOS Global Public Health’s publication criteria as it currently stands. Therefore, we invite you to submit a revised version of the manuscript that addresses the points raised during the review process.

You will see that both reviewers were positive about this version but both gave some additional (very concrete and hopefully straightforward to address) feedback. I agree with the reviewers that this is close to publication ready and will look forward to receiving your revised manuscript.

A letter that responds to each point raised by the editor and reviewer(s). You should upload this letter as a separate file labeled ’Response to Reviewers’.A marked-up copy of your manuscript that highlights changes made to the original version. You should upload this as a separate file labeled ’Revised Manuscript with Track Changes’.An unmarked version of your revised paper without tracked changes. You should upload this as a separate file labeled ’Manuscript’.

We look forward to receiving your revised manuscript.

Kind regards,

Nicola L. Hawley

Academic Editor

Journal Requirements:

Additional Editor Comments (if provided):

Reviewers’ comments:

Reviewer’s Responses to Questions

**Comments to the Author**

1. If the authors have adequately addressed your comments raised in a previous round of review and you feel that this manuscript is now acceptable for publication, you may indicate that here to bypass the “Comments to the Author” section, enter your conflict of interest statement in the “Confidential to Editor” section, and submit your "Accept" recommendation.

Reviewer #1: (No Response)

Reviewer #2: All comments have been addressed

2. Does this manuscript meet PLOS Global Public Health’s publication criteria? Is the manuscript technically sound, and do the data support the conclusions? The manuscript must describe methodologically and ethically rigorous research with conclusions that are appropriately drawn based on the data presented.

Reviewer #1: Yes

Reviewer #2: Yes

3. Has the statistical analysis been performed appropriately and rigorously?

Reviewer #1: Yes

Reviewer #2: Yes

4. Have the authors made all data underlying the findings in their manuscript fully available (please refer to the Data Availability Statement at the start of the manuscript PDF file)?

Reviewer #1: No

Reviewer #2: Yes

5. Is the manuscript presented in an intelligible fashion and written in standard English?

Reviewer #1: Yes

Reviewer #2: Yes

6. Review Comments to the Author

Reviewer #1: Thank you for addressing the majority of reviewer comments and for responding to each reviewer’s comments. There are a few topics that I still believe require some minor revisions:

-I see that you site the parent study once in the introduction, but I would cite it again in the methods to make it easier for readers to reference.

-I think your move to only have two aims does improve clarity and the aims are stated clearly in 89-94. However, there is still some confusion in how these aims are discussed throughout the paper. For example, I *think* the first few paragraphs of the results address your first aim. However, Table 2 is referenced and is located in this section and Table 2 seems to provide results related to Aim 2.

-The results for Aim 2 are very clear and comprehensive, but I am not sure what the qualitative results are for Aim 1. Explicit themes are not highlighted. I would include a subheading at the start of the results section to reference acceptability of group care, and then clearly present themes and any quantitative data you have to support acceptability. So inclusion and discussion of Table 3, which I think goes with your first aim, would be moved up. Then you can move to discussion of Aim 1 qual and quant results

Reviewer #2: Thank you for the opportunity to review this manuscript - it does add value add to the literature and is is very close to ready. I appreciate how responsive the authors were to the reviewers.

Minor suggestions:

1. The theory of change is discussed briefly in the introduction, but it really is a guiding principle according the 2020 article as well as the actual theory of change article. It is considered on the causal pathway. A touch more about how this is foundational to the present study and results is needed since, it seems, this the first to actually pull in self-efficacy. I appreciate how self-efficacy, through the TOC, is directly linked into table 2 and revisited in the discussion. The introduction could use a touch more positioning of the 2020 and TOC articles since these are foundational and at the core of this paper (Appendix 4 is great and really helpful for context. Suggestion: Number the appendices in order of presentation in the manuscript; i.e., Appendix 4 is the first one cited in the introduction. At the same time, Bandura is overexplained and could be tightened up to improve flow and overall reading.

2. Typo/grammar: Line 102 left an incomplete phrase, "Every woman was" and the sentence on Line 104 - "interacted as a group" is odd phrasing give it is a participatory session.

3. Table 2 is nice, but since it seems it includes the same quotes that are in the text, so then it feels redundant. Given multiple speakers are presented in column 3, it is nearly unreadable. The data are not useful to readers in this format. Illustrative quote tables need to provide information that the text does not; so it seems it could be deleted since no new information is coming from it.

4. Domain 1: Emergency care-seeking: The first two illustrative quotes do not seem to fit with an emergency decision, but rather about timing and going to a health facility (domain #4). In part, the final analysis to group into domains feels incomplete or forced for the emergency domain (the third quote perfectly aligns). Are there more to the quotes to show its referring to emergencies arising because as written just seems to be about going into labor and not an emergency situation.

4. Domain 2: The Pregnancy Management domain quotes are difficult for readers unfamiliar with cultural aspect of the leaf water, so it is unclear what the person is saying about the leaf water as self-care.

5. Domain 4: Facility Delivery: These quotes seem like they are related to birth preparedness (see Ghana studies) and also advocacy aspects related to when to go to a facility. It seems that quotes can fall into multiple categories, but then it makes the domain names feel slightly off. Domains 1 and 4 seem to capture seeking care.

6. Lines 240-242, vicarious: the selected quote does not really show that there is learning from others experiences at work.

7. I assume formatting of tables and appendices will happen at the copy edit phase.

8. Discussion - linkages of domains and concepts to other published results like was done for IPTp is needed to connect to other studies in LMICs (e.g., reflecting knowledge change, empowerment, birth preparedness and complication readiness, preference for G-ANC, etc.) would strengthen the self-efficacy contribution. It helps with the logic argument because it would show that the innovative analysis and results from Benin, resonate with other studies that had similar outcomes but did not frame this as self-efficacy. In other words, there is strong argument that this first article to approach it in this way, contribute general knowledge by labeling it as such.

Overall, this is an improved version and nearly ready for publication. The authors were attentive to reviewers concerns.

7. PLOS authors have the option to publish the peer review history of their article (what does this mean?). If published, this will include your full peer review and any attached files.

**Do you want your identity to be public for this peer review?** For information about this choice, including consent withdrawal, please see our Privacy Policy.

Reviewer #1: No

Reviewer #2: No

 Figure Resubmissions:

---

## [Decision Letter · Decision Letter 2]

7 Apr 2026

“Each moon we come to weigh the pregnancy:” Exploring the experience of group antenatal care processes in Benin and their contributions to self-efficacy

PGPH-D-25-01395R2

Dear Professor Winch,

We are pleased to inform you that your manuscript ’“Each moon we come to weigh the pregnancy:” Exploring the experience of group antenatal care processes in Benin and their contributions to self-efficacy’ has been provisionally accepted for publication in PLOS Global Public Health.

If your institution or institutions have a press office, please notify them about your upcoming paper to help maximize its impact. If they’ll be preparing press materials, please inform our press team as soon as possible -- no later than 48 hours after receiving the formal acceptance. Your manuscript will remain under strict press embargo until 2 pm Eastern Time on the date of publication. For more information, please contact globalpubhealth@plos.org.

Best regards,

Nicola L. Hawley

Academic Editor

Reviewer Comments (if any, and for reference):

Reviewer’s Responses to Questions

**Comments to the Author**

1. If the authors have adequately addressed your comments raised in a previous round of review and you feel that this manuscript is now acceptable for publication, you may indicate that here to bypass the “Comments to the Author” section, enter your conflict of interest statement in the “Confidential to Editor” section, and submit your "Accept" recommendation.

Reviewer #1: All comments have been addressed

Reviewer #2: All comments have been addressed

2. Does this manuscript meet PLOS Global Public Health’s publication criteria? Is the manuscript technically sound, and do the data support the conclusions? The manuscript must describe methodologically and ethically rigorous research with conclusions that are appropriately drawn based on the data presented.

Reviewer #1: Yes

Reviewer #2: Yes

3. Has the statistical analysis been performed appropriately and rigorously?

Reviewer #1: Yes

Reviewer #2: Yes

4. Have the authors made all data underlying the findings in their manuscript fully available (please refer to the Data Availability Statement at the start of the manuscript PDF file)?

Reviewer #1: Yes

Reviewer #2: Yes

5. Is the manuscript presented in an intelligible fashion and written in standard English?

Reviewer #1: Yes

Reviewer #2: Yes

6. Review Comments to the Author

Reviewer #1: Great job! This will be a wonderful value add to the literature.

Reviewer #2: I thank the authors for their thoughtful responses to previous reviews.

7. PLOS authors have the option to publish the peer review history of their article (what does this mean?). If published, this will include your full peer review and any attached files.

**Do you want your identity to be public for this peer review?** For information about this choice, including consent withdrawal, please see our Privacy Policy.

Reviewer #1: No

Reviewer #2: No
